# Zebrafish-Based Screening Models for the Identification of Anti-Metastatic Drugs

**DOI:** 10.3390/molecules25102407

**Published:** 2020-05-21

**Authors:** Joji Nakayama, Hideki Makinoshima

**Affiliations:** 1Shonai Regional Industry Promotion Center, Tsuruoka, Yamagata 997-0052, Japan; 2Tsuruoka Metabolomics Laboratory, National Cancer Center, Mizukami 246-2, Kakuganji, Tsuruoka, Yamagata 975-0052, Japan; hmakinos@east.ncc.go.jp; 3Division of Translational Research, Exploratory Oncology Research, and Clinical Trial Center, National Cancer Center, Kashiwa, Chiba 277-8577, Japan

**Keywords:** zebrafish, metastasis, EMT, angiogenesis, phenotyping screening

## Abstract

Metastasis, a leading contributor to the morbidity of cancer patients, occurs through a multi-step process: invasion, intravasation, extravasation, colonization, and metastatic tumor formation. Each process is not only promoted by cancer cells themselves but is also affected by their microenvironment. Given this complexity, drug discovery for anti-metastatic drugs must consider the interaction between cancer cells and their microenvironments. The zebrafish is a suitable vertebrate animal model for in vivo high-throughput screening studies with physiological relevance to humans. This review covers the zebrafish model used to identify anti-metastatic drugs.

## 1. Introduction

Overt metastases, the end result of malignant alteration in cancer cells, are responsible for approximately 90% of cancer-associated mortality. Metastasis consists of a multi-step process: invasion in which neoplastic epithelial cells invade into the adjacent tissue after they lose cell-cell adhesion; intravasation in which tumor cells penetrate through the endothelium of blood or lymphatic vessels to enter the systemic circulation; survival in the circulatory system in which certain circulating tumor cells appear able to survive in the bloodstream; extravasation in which cancer cells extravasate through the capillary endothelium at distal sites; colonization in which cancer cells proliferate in a new environment, and metastatic tumor formation in which cancer cells form a new tumor in secondary organs (Figure 1) [1,2,3]. Recent studies demonstrated that each of these processes is promoted not only by cancer cells but also by the tumor microenvironment [4,5]. Therefore, the interaction between cancer cells with their microenvironments must be considered while developing therapeutic strategies for metastasis.

From the therapeutic point of view, the metastatic process can be divided into two phases: the early phase including tumor cell invasion and extravasation, and the late phase consisting of colonization and metastatic tumor formation. During the former phase, anti-metastatic treatments focus on inhibiting the spread of cancer cells to distant organs. Likewise, the goal of treatment during the latter phase is to inhibit the proliferation of disseminated cancer cells in distant organs [6].

Cancer research using zebrafish as a model has attracted attention because this model offers many unique advantages that are not readily provided by other animal models [7,8,9]. For example, (i) zebrafish have orthologues to 86% of 1318 human drug targets, 71% of human proteins, and 82% of disease-causing human proteins [10,11]; (ii) drugs are administered to zebrafish by dissolving them in water, except hydrophobic drugs that are administered to the zebrafish by microinjection [12,13]; (iii) the effects of chemicals may be evaluated through direct observation due to the transparency of zebrafish embryos; (iv) zebrafish generate large numbers of progeny, thus increasing the power of statistical analyses; (v) husbandry expenses associated with zebrafish are much lower than for mammals due to their low-maintenance nature. These advantages have made zebrafish a popular platform for drug screening.

However, there are several limitations associated with zebrafish. One limitation of zebrafish is that they are not mammals. Therefore, the effects of the drugs observed in zebrafish models must be validated in advanced preclinical models. Another limitation of using zebrafish for in vivo drug screening is that it is difficult to test drugs that are insoluble in water. A previous study that screened 23 drugs known to cause cardiotoxicity in humans yielded four out of five false-negative results in zebrafish due to poor absorption. The efficacy of drugs that yielded false negatives was confirmed by microinjection studies [13]. The other limitation associated with zebrafish is that the optimum temperature for breeding zebrafish is 28 °C; this temperature differs from that of the human body by 9 °C (37 °C). In xenograft experiments that involve the transplantation of human or mouse cells into zebrafish, the zebrafish maintained a temperature of 31–34 °C, since 28 °C is not optimal for the growth of the transplanted cells. However, previous studies demonstrate that human cancer cells that are inoculated into zebrafish embryos have a better proliferation index at 36 °C than at 34 °C. Therefore, 36 °C is considered the most suitable temperature for testing chemotherapeutic drugs such as 5-fluorouracil. Although the zebrafish is a poikilothermic and eurythermal animal, there was a 10% increase in the mortality rate of zebrafish maintained at 36 °C compared to that of zebrafish at 34 °C [14].

Several anti-metastatic drugs have been identified through studies using zebrafish models [15]. The transplantation of cancer cells into zebrafish embryos is most frequently used in these models. Briefly, human cancer cells labeled with a fluorescent marker such as enhanced green fluorescent protein (EGFP) are inoculated into larvae or embryos of zebrafish. This is because zebrafish embryos are transparent and do not have a fully developed adaptive immune system until 21 days post fertilization (dpf) [16]. Zebrafish are maintained for a few days with or without the drug. The effects of the drug are evaluated by observing the behavior of the cell in the zebrafish during tumor invasion, migration, extravasation, and angiogenesis (Figure 2). Drugs that have been shown to suppress metastatic processes in zebrafish xenograft models are listed in Table 1.

Another approach that is frequently used to assess anti-metastatic drugs is phenotype-based screening. Phenotype-based screening has been increasingly employed in biomedical and pharmaceutical research since the contributions of phenotypic screening to the discovery of first-in-class small-molecule drugs exceeded that of target-based approaches [31]. Diverse zebrafish-based screening experiments with various phenotypic readouts are conducted, which results in the identification of candidate substances that are further examined in preclinical and clinical trials [32,33]. The drugs that have been shown to suppress metastasis through phenotype-based screening models are listed in Table 2.

This review focuses on zebrafish models that have been used as screening platforms for the identification of anti-metastatic drugs. Figure 1 illustrates which of the identified drugs suppress each step of metastasis.

## 2. Targeting Metastatic Dissemination

Metastatic dissemination is the initial step of metastasis. During this step, cancer cells in the primary tumor site invade the neighboring tissue, enter the blood vessels, survive in the circulation, and exit the vessels to penetrate foreign tissues. Different genes and molecular mechanisms are responsible for each of these steps [40,41].

Integrins are cell-surface adhesion receptors consisting of α and β transmembrane protein subunits, which directly interact with extracellular matrix (ECM) components. Integrins also contribute to tumor development and metastasis progression [42]. The αv integrin antagonist GLPG0187 suppresses the metastatic dissemination of breast cancer cells in a zebrafish xenograft model that involves the inoculation of mCherry-labeled MDA-MB-231 cells into the duct of Cuvier of *Tg(fli1: EGFP)*, which expresses EGFP in all blood vessels [43]. In this study, 56% of vehicle-treated zebrafish developed aggressive tumor lesions on their tail fins. In contrast, only 18% of GLPG0187-treated fish developed secondary tumors on the tail fin. Genetic inhibition of αv integrin shows the same effect as the pharmacological inhibition with GLPG0187, thus demonstrating that the anti-metastatic effects of GLPG0187 result from the inhibition of αv integrin rather than its off-target effects [17]. 

C-X-C chemokine receptor type 4 (CXCR4) and its cognate ligand C-X-C motif chemokine 12 (CXC12), which is also known as stromal cell-derived factor 1, play a pivotal role in regulating physiological processes such as hematopoiesis, leukocyte trafficking and cell migration [44]. The binding of CXC12 to CXCR4 also contributes to tumor development and metastatic progression [45]. Furthermore, CXCR4-overexpressing cancer cells preferentially develop metastatic tumors in distant organs that produce high levels of CXCL12 in human specimens and murine models [46]. The same phenomenon was also confirmed in a zebrafish xenograft model; in this model, a sub-cell line of MDA-MB-231 cells that overexpresses CXCR4 (MDA-MB-231-B) showed more aggressive metastatic behavior compared to the parental MDA-MB-231 cells upon inoculation into the duct of Cuvier of *Tg(kdrl:EGFP)*, which selectively expresses EGFP in endothelial cells. However, MDA-MB-231-B cells that are inoculated into *cxcl12a^−/−^/cxcl12b^−/−^* background *Tg(kdrl:EGFP)* zebrafish fail to induce invasion and the formation of micro-metastases in the tail fin. Moreover, pharmacological inhibition using IT1t, a potent CXCR4 antagonist, suppresses early metastatic events of the cells *in vivo*. Genetic inhibition of CXCR4 shows the same effect as treatment with IT1t [18].

Members of the proto-oncogenic Src family of non-receptor protein tyrosine kinases also play critical roles in several metastatic cellular signal transduction pathways. The activation of Src family kinases in human cancers may occur through a variety of mechanisms, and the activity of Src family kinases is often critical for tumor progression. Several studies demonstrate that Src-mediated signaling plays a critical role in promoting metastasis [47,48]. A phenotype screening study using *Tg(cldnb: EGFP)* to mark the migrating posterior lateral line primordium as a readout for inhibition of collective cell migration identified the Src kinase inhibitor SU6656 has as an anti-metastatic agent. A murine model of metastatic cancer using 4T1 murine mammary carcinoma cells confirms that SU6656-treated mice show significantly fewer surface metastases in the lung compared with vehicle-treated mice [34].

Besides the three aforementioned drugs, several other pharmacological agents suppress the metastatic dissemination of human cancer cells in zebrafish xenograft models. The transforming growth factor-β (TGF-β receptor inhibitors SB431542 or SB525334 suppress the metastatic dissemination of breast cancer cells or undifferentiated pleomorphic sarcoma (UPS) cells in zebrafish xenograft models, respectively [19,20]. The inhibition of the phosphoinositide-dependent kinase-1 (PDK1)/Phospholipase C gamma 1 (PLCγ1) complex with the small molecule inhibitor 2-O-Bn-InsP5 significantly reduces the metastatic dissemination of MDA-MB-231 cells in *Tg(kdrl: HsHRAS- mCherry)*^s896^ zebrafish embryos [21]. The sirtuin (SIRT) 1/2 inhibitor tenovin-6 suppresses the metastatic dissemination of TC252 Ewing sarcoma cells in *Tg(fli1: EGFP)* [22]. Other compounds that suppress the metastatic dissemination of cancer cells in zebrafish xenograft models are listed in Table 1.

Experimental studies demonstrate that cancer cells can disseminate systemically from the earliest epithelial alterations in *HER-2* and *PyMT* transgenic mice [49]. Clinical studies also reveal that cancer cells disseminate during the earliest phase of metastasis and are detected in the bone marrow years before the development of overt metastases [50]. These findings suggest that targeting the metastatic dissemination of cancer is not an effective strategy for blocking metastasis. However, the metastatic dissemination of cancer cells is directly observed in the living zebrafish due to the transparency of zebrafish embryos. This is a unique advantage possessed by zebrafish models. Combining this advantage with new imaging technologies would allow for the identification of molecular mechanisms responsible for the metastatic dissemination of cancer cells. The insights yielded by these future findings may lead to the discovery of new anti-metastatic drugs.

## 3. Targeting the Epithelial-Mesenchymal Transition (EMT) Process

EMT plays a central role in early embryonic morphogenesis, its program enables various types of epithelial cells to convert into mesenchymal cells [51]. Experimental studies demonstrate that EMT also contributes to metastatic progression by increasing the invasiveness, motility, and resistance of cancer cells to chemotherapy/apoptosis [52]. Therefore, EMT would be an ideal therapeutic target for anti-metastatic drugs.

Two transgenic zebrafish models that offer a screening platform for the identification of anti-EMT drugs have been reported. One is a tamoxifen-controllable *Twist1a-ER^T2^* transgenic zebrafish line *Tg(fabp10a:mCherry-T2A-Twist1a-ER^T2^)*, which induces spontaneous metastatic dissemination of cancer cells through induction of EMT. The other is the *snailb* promoter-driven GFP transgenic zebrafish line that labels epithelial cells undergoing EMT in zebrafish embryos [23,35].

A tamoxifen-controllable *Twist1a-ER^T2^* transgenic zebrafish serves as a platform for the discovery of anti-metastatic drugs. The activation of Twist1a-ER^T2^ following 48 h of tamoxifen treatment induces the conversion of epithelial cells into mesenchymal cells in the liver. By crossing this model with *xmrk* (a homolog of the hyperactive form of EGFR) transgenic zebrafish [53], which develops hepatocellular carcinoma, approximately 80% of the double-transgenic zebrafish showed spontaneous dissemination of mCherry-labeled hepatocytes from the liver to the entire abdomen region and the tail region within five days from the treatment initiation. FDA-approved drugs are subjected to *in vivo* screening using this model. Adrenosterone, an inhibitor of hydroxysteroid (11-beta) dehydrogenase 1 (HSD11β1), suppresses cellular dissemination in this model (Figure 3). This suppressor effect is validated in a zebrafish xenotransplantation model in which highly-metastatic human cell lines are inoculated into the duct of Cuvier of *Tg(kdrl:EGFP)* transgenic zebrafish. Genetic inhibition of HSD11β1 also suppresses the metastatic dissemination of these cell lines in a zebrafish xenotransplantation model. This suppression results from the re-expression of E-cadherin and other epithelial markers and lost partial expression of mesenchymal markers through the down-regulation of Snail and Slug [23].

*Tg(snailb: GFP)*, which provides a whole-animal EMT reporter system in zebrafish, is established for rapid drug screening. This model allows for the labeling of epithelial cells undergoing EMT to produce sox10-positive neural crest (NC) cells. Treating embryos of this model with candidate small-molecule EMT-inhibiting compounds previously demonstrated that TP-0903, an inhibitor of AXL receptor tyrosine kinase, blocked the delamination of cranial NC cells in both the lateral and medial populations. TP-0903 stimulates retinoic acid (RA) biosynthesis and RA-dependent transcription. These studies identified TP-0903 as an activator of RA *in vivo* and raised the possibility that its prior success in eliminating disseminated cancer cells depends upon the RA-dependent inhibition of EMT [35].

The activation of EMT endows epithelial carcinoma cells with mesenchymal traits, stem-like characteristics, increased drug resistance, invasiveness, and metastatic ability [54]. Furthermore, a recent study shows that EMT needs to be transient and reversible, and the transition from a fully mesenchymal phenotype to an epithelial-mesenchymal hybrid state or a fully epithelial phenotype is associated with malignant phenotypes [55]. Cortisol, which is produced by HSD11β1, is reported to promote the metastatic progression of breast cancer cells. The knockdown of the cortisol receptor in highly metastatic cells prevents them from metastasizing to distant organs [56]. TP-0903 is also reported to sensitize erlotinib-resistant non-small cell lung cancer cells to erlotinib by reversing the mesenchymal phenotype in preclinical tumor models [57]. This evidence supports that adrenosterone and TP-0903 would suppress the metastatic progression of human cancer cells.

## 4. Targeting Angiogenesis

Angiogenesis is the formation and remodeling of new blood vessels and capillaries from pre-existing blood vessels. The growth potential of avascular tumors is severely restricted due to the lack of blood supply. The formation of new blood vessels provides avascular tumors with nutrients, oxygen, and more efficient removal of waste products [58]. Although anti-angiogenic agents targeting vascular endothelial growth factor (VEGF) signaling are already used to effectively treat some cancers, there is a continued need for the development of new inhibitors of angiogenesis to circumvent resistance or reduce toxicity.

A pioneering study demonstrates that the treatment of zebrafish embryos with the VEGF receptor inhibitor PTK787/ZK222584 completely blocks the formation of all major blood vessels. Furthermore, the overexpression of the downstream effector AKT/PKB allows blood vessels to form in the presence of the drug [36]. Following this study, several studies performing high-throughput screening in zebrafish models are reported to identify new anti-angiogenic drugs [37].

A transgenic zebrafish that expresses green reef coral fluorescent protein (GRCFP) under the control of the *VEGFR2* promoter Tg(VEGFR2: GRCFP) offers a high-throughput platform for the discovery of anti-angiogenic drugs. Previous *in vivo* drug screening studies using this model identified indirubin-3′-monoxime (IRO) as an inhibitor of angiogenesis that did not affect vasculogenic vessel development or preexisting vasculature. An *in vitro* assay using human umbilical vein endothelial cells (HUVECs) confirmed that IRO inhibits two major components of the angiogenic process—endothelial tube formation and cell proliferation. However, IRO does not significantly affect endothelial cell migration [37]. Another high-throughput screening study using *Tg(fli: EGFP)* also identified the PhK subunit G1 (PhKG1) inhibitor F11 as an anti-angiogenic drug *in vivo* and *in vitro* using cultured human endothelial cells [38]. 

A previous study using high-resolution confocal microscopy allows one to visualize the neovascularization and behavior of highly metastatic cells. In the study, the DsRed-labeled MDA-MB-435 human melanoma cell line was inoculated into the peritoneal cavity of *Tg(fli1: EGFP)* zebrafish at 2 dpf. The interactions between the DsRed-labeled cells and the GFP-labeled vessels during angiogenesis are visualized through three-dimensional reconstructions of the interaction. The images reveal that RhoC amplification in the DsRed cells induces the formation of dynamic membrane protrusions and blebs and that VEGF secreted by the DsRed cells increases vascular permeability. These molecules work cooperatively to facilitate the invasion and intravasation by regulating the cytoskeletal and vascular remodeling. Pharmacological inhibition of VEGFR using SU5416 restored the integrity of the vessel wall and decreased the size of the DsRed tumor from 25.5 to 16 μm [24].

Another zebrafish model of metastasis may be used to visualize cancer cell dissemination, invasion, and metastasis at the single-cell level. Using this model, DiI-labeled T241 murine fibrosarcoma cells are injected into the perivitelline space of *Tg(fli1: EGFP)* zebrafish at 48 hpf and the zebrafish is subsequently transferred to a hypoxia chamber. Under hypoxia, the cells disseminate from their primary sites, invade the neighboring tissue, and metastasize to distal parts of the body through the upregulation of VEGF, which is a hypoxia-regulated angiogenic factor. In contrast, significant dissemination of cells does not occur under normoxia. Sunitinib, a VEGF inhibitor, inhibits hypoxia-induced invasion, dissemination, and metastasis of T241 tumors in this model [25]. The other compounds that suppress angiogenesis are listed in Table 2.

The U.S. Food and Drug Administration (FDA) has approved several anti-angiogenic drugs to treat cancer. Most of the inhibitors target VEGF, its receptor, or other specific molecules involved in angiogenesis [59]. Among the drugs verified by zebrafish models, only sunitinib has been approved by the FDA. PTK787/ZK222584 (also known as vatalanib) has been extensively tested in Phase I, II, and III clinical trials [60,61,62]. Clinical trials for SU5416 were discontinued due to discouraging results [63,64,65]. The limited efficacy of anti-angiogenic drugs remains a continuing problem in their clinical use [66,67]. Unique advantages provided by zebrafish models that would be suitable for angiogenesis research including the discovery and evaluation of anti-angiogenic agents, the identification of novel targets for anti-angiogenesis, and the elucidation of molecular mechanisms responsible for angiogenesis. Therefore, new insights from these studies would contribute to improving the therapeutic benefits of anti-angiogenic strategies.

## 5. Targeting Lymphangiogenesis

The growth of new lymphatic vessels (lymphangiogenesis) in tumors is an integral part of metastasis. Cancer cells in the primary tumor site initially spread via lymphatic vessels to their regional lymph nodes [68]. Previous *in vivo* high-throughput screening studies using *Tg(lyve1: EGFP)*, which expresses EGFP in the lymphatic vessels [69], identified that treatment with kaempferol, which inhibits VEGFR2/3, suppresses the migration of lymphatic precursors known as secondary sprouts from the posterior cardinal vein. Further, a murine *in vivo* lymphangiogenesis Matrigel plug assay confirmed that kaempferol has anti-lymphatic activity. A mouse xenograft model where luciferase-expressing MDA-MB-231 cells (MDA-MB-231-luc-D3H2LN) were inoculated into the mammary fat pad of nude mice demonstrated that kaempferol reduced lymph node metastases compared with vehicle treatment. However, bioluminescence imaging and hematoxylin and eosin (H&E) staining showed that 80% of kaempferol-treated mice had metastatic tumors in the pancreas and diaphragm in both the prevention and intervention regimens, compared with 10% of control mice. These results indicate that kaempferol is only effective at reducing lymph node metastases [39].

Kaempferol is widely distributed in the plant kingdom and is a common constituent of fruits and vegetables. Kaempferol and its derivatives have cardioprotective, neuroprotective, anti-inflammatory, antidiabetic, antioxidant, antimicrobial properties [70]. Past studies demonstrate that kaempferol suppresses the proliferation of various cancer cells derived from the lung, breast, pancreas, prostate, and colon [71]. Moreover, multi-ethnic cohort studies have demonstrated that flavonols such as kaempferol prevent the development of pancreatic cancer [72]. Therefore, the daily intake of kaempferol may prevent the progression of metastasis.

## 6. Targeting Tumor Microenvironments

Cancer cells are surrounded by many different types of cells: vascular endothelial cells, immune cells, fibroblasts, adipocytes, and the extracellular matrix (ECM). Experimental studies demonstrate that crosstalk between cancer cells and the surrounding microenvironment promotes metastasis. Therefore, targeting the tumor microenvironment may be an effective therapeutic strategy for suppressing metastasis [73].

The highly metastatic zebrafish melanoma cell line, ZMEL1 is established from melanomas of *Tg(mitfa: BRAF^V600E^; p53^−/−^)* fish, which develop melanoma by four months old [74]. After subcutaneously inoculating the cells into the ventral flank of adult *casper* recipient zebrafish, 83% (*n* = 25/30) of the fish showed widespread spontaneous metastases in the anterior region just behind the gill structure. Other metastases were observed on the posterior tail musculature and eye two weeks after the inoculation [75]. Studies using this model show that the interaction between disseminated melanoma cells and their microenvironment plays a critical role in promoting the formation of metastatic tumors at distant sites. Adipocytes increase the proliferation and invasion of adjacent ZMEL1 cells by transferring lipids to the cells. The lipids are incorporated into the cells via the fatty acid transport protein (FATP)/SLC27A family of lipid transporters, which are overexpressed by the cells. Pharmacological inhibition of FATPs with the small-molecule inhibitor lipofermata reduces the proliferation and invasion of ZMEL1 cells by interfering with lipid transport [27]. Also, endothelin-3 (EDN3), which is expressed in the microenvironment surrounding ZMEL1 cells, promotes a differentiated and proliferative state through the up-regulation of MYC, premelanosome protein (PMEL), tyrosinase-related protein 1 (TYRP1), and tyrosinase (TRY). The ZMEL1 cells that are transplanted into EDN3-deficient zebrafish decrease the formation of metastatic tumors compared to those in the wild-type fish [76]. 

Due to the phenotypic plasticity of cancer cells that emerge from interactions between cancer cells and their microenvironment, cancer cells have developed resistance to anti-cancer drugs, which have been identified from *in vitro* screens using two-dimensional cell culture systems. Therefore, drug discovery needs to also consider the phenotypic plasticity of cancer cells. Zebrafish models can be used for *in vivo* drug screening to evaluate the effect of chemicals on cancer cells while accounting for the tumor microenvironment. Lipofermata, which targets the tumor microenvironment, is one example of an anti-metastatic drug that has been identified by using a zebrafish model [27]. A recent study also reports that the pharmacological inhibition of FATP2 with lipofermata abrogates the activity of pathologically-activated neutrophils, which play a critical role in regulating the immune response during cancer progression, and substantially delays tumor progression [77]. This evidence supports that targeting the microenvironments that surround cancer cells would be an effective strategy for blocking metastasis.

## Figures and Tables

**Figure 1 molecules-25-02407-f001:**
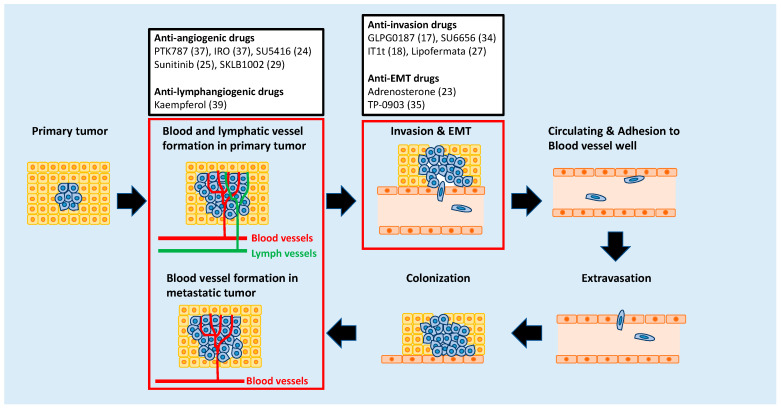
Anti-metastatic drugs identified through zebrafish-based screening that target different stages of metastasis. The numbers in parentheses indicate reference.

**Figure 2 molecules-25-02407-f002:**
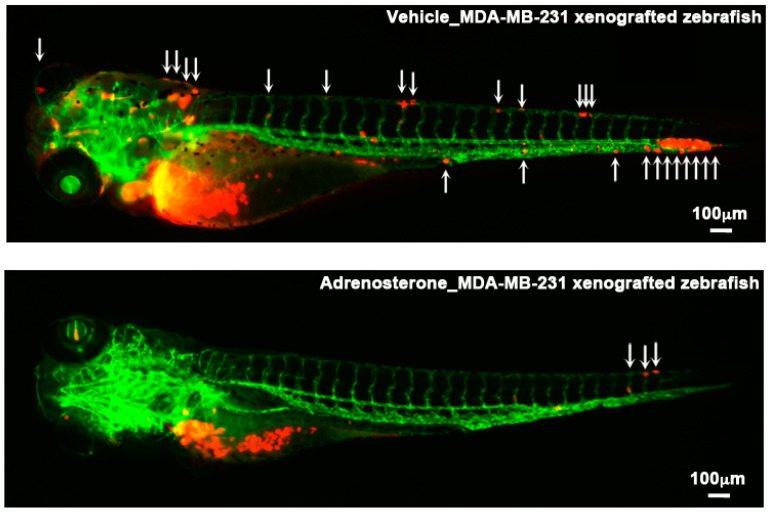
Examples of zebrafish xenograft models. Metastatic dissemination of red fluorescence protein (RFP)-labeled MDA-MB-231 cells in the vehicle-(top) or adrenosterone (bottom)-treated *Tg(kdrl: EGFP)* fish. The cells are inoculated into the duct of Cuvier of the fish at 48 hpf and then treated with either vehicle or adrenosterone for 24 h. White arrows indicate disseminated MDA-MB-231 cells. Images are shown in 4× magnification. Scale bar = 100 μm. Images are reprinted from [23].

**Figure 3 molecules-25-02407-f003:**
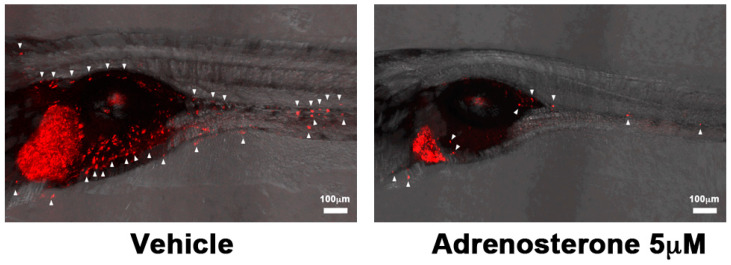
Examples of phenotype-based screening in zebrafish. Metastatic dissemination of mCherry-positive cells from the liver of vehicle (left) or adrenosterone (right)-treated *Twist1-ER^T2^/xmrk* double-transgenic fish. White arrows indicate disseminated mCherry-positive cells. Images are shown as Z-stack images using 100× magnification. Scale bar = 100 μm. Images are reprinted from [23].

**Table 1 molecules-25-02407-t001:** A list of the anti-metastatic drugs identified in zebrafish xenograft models.

Drug	Targeting Molecule	Targeting Molecular Event	Cancer Cell	Recipient Zebrafish	Inoculation Site	Reference
GLPG0187	αv integrin	Invasion	MDA-MB-231 (Breast)	*Tg*(*Fli1: EGFP*)	Duct of Cuvier	[17]
IT1t	CXCR4	Invasion	MDA-MB-231 (Breast)	*Tg*(*Fli1: EGFP*)	Duct of Cuvier	[18]
SB431542	TGFβR1	Invasion	MDA-MB-231 (Breast)	*Tg*(*Fli1: EGFP*)	Duct of Cuvier	[19]
SB525334	TGFβR1	Invasion	KIA (UPS)	*Tg*(*Fli1: EGFP*)	Yolk sac	[20]
2-O-Bn-InsP5	PDK1	Invasion	MDA-MB-231 (Breast)	*Tg*(*kdrl: HsHRAS-mCherry*)*^s896^*	Duct of Cuvier	[21]
Tenovin-6	SIRT1/2	Migration	TC252 (Ewing sarcoma) and A673 (Ewing sarcoma)	*Tg*(*Fli1: EGFP*)	Duct of Cuvier	[22]
Adrenosterone	HSD11β1	EMT	HCCLM3 (Liver) and MDA-MB-231 (Breast)	*Tg*(*kdrl: EGFP*)	Duct of Cuvier	[23]
SU5416	VEGFR	Angiogenesis	MDA-MB-435 (Skin)	*Tg*(*Fli1: EGFP*)	Peritoneal cavity	[24]
Sunitinib	VEGF	Angiogenesis	T241 (Thyroid)	*Tg*(*Fli1: EGFP)*	Perivitelline space	[25]
Sunitinib	VEGF	Invasion	SJmRBL-8 (Retinoblastoma)	*Tg*(*Fli1: EGFP*)	Eye	[26]
LY294002	PI3-kinase	Invasion	MDA-MB-231 (Breast)	*Tg*(*Fli1: EGFP*)	Duct of Cuvier	[19]
GM6001	MMPs	Invasion	MDA-MB-231 (Breast)	*Tg*(*Fli1: EGFP*)	Duct of Cuvier	[19]
Lipofermata	FATP	Invasion	ZMEL1 (Skin)	*Casper*	Subcutaneous tissue	[27]
DHS	Unknown	Invasion	LLC (Lung)	*Tg*(*Fli1: EGFP*)	Perivitelline cavity	[28]
SKLB1002	VEGFR2	Angiogenesis	B16-F10 (Skin)	*Tg*(*Fli1: EGFP)*	Perivitelline space	[29]
Osimertinib	EGFR	Angiogenesis	H1975 (Lung)	*Tg*(*Fli1: EGFP*)	Duct of Cuvier	[30]

**Table 2 molecules-25-02407-t002:** A list of the anti-metastatic drugs identified through phenotype-based screening using zebrafish.

Drug	Targeting Molecule	Targeting Molecular Event	Screening Platform	# of Drugs Subjected to the Screen	Reference
SU6656	Src	Migration	*Tg*(*cldnb: EGFP*)	2960 compounds from the LOPAC1280 library, the NatProd library, and the GSK Published Kinase Inhibitor Set (PKIS)	[34]
Adrenosterone	HSD11β1	EMT	*Tg*(*Fabp10A: mCherry-T2A-Twist1-ER^T2^*) × *Tg*(*fabp10a: TA; TRE:xmrk; krt4: GFP*)	68 chemicals from the Prestwick Chemical Library	[23]
TP-0903	AXL receptor	EMT	*Tg*(*snai1b: GFP*)	Not applicable	[35]
PTK787	VEGFR	Angiogenesis	WT	Not applicable	[36]
IRO	Unknown	Angiogenesis	*Tg*(*VEGFR2: GRCFP*)	1280 chemicals from the LOPAC1280 library	[37]
F11	PhKG1	Angiogenesis	*Tg* (*fli1: EGFP*)	288 chemicals	[38]
Kaempferol	VEGFR2/3	Lymphangiogenesis	*Tg*(*lyve1: EGFP*)	1120 chemicals from the Prestwick Chemical Library	[39]

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
