# Peer review of "Zebrafish-Based Screening Models for the Identification of Anti-Metastatic Drugs"

_molecules, 2020, doi:10.3390/molecules25102407_

Round 1

Reviewer 1 Report

The review “Zebrafish-based screening platform for identification of an anti-metastasis drug” focuses on different zebrafish xenograft models and phenotype screening approaches, which were employed to identify inhibitors of processes involved in metastasis.

After a short introduction explaining the different processes involved in metastasis as well as the zebrafish xenograft models and the phenotype-based screening, the review is partitioned into subsections concerning the different target processes (metastatic dissemination, EMT, angiogenesis, tumor microenvironments).

I think that the topic of the review is highly interesting to the readers of the journal since the use of zebrafish models for drug discovery is a widely expanding area of research.

In general, I kindly suggest that the authors make use of an English editing service or find a native speaker, that can assist in improving the language of the manuscript. In most parts, the language is understandable but contains a lot of errors, in some parts it is simply confusing. I am not a native speaker myself and I don’t feel comfortable to revise the language, I just would like to mention a few examples:

  • In general: discovery instead of discoveries (most of the times..)
  • Line 38: the end result
  • Line 43: add “is” after …themselves but…
  • Line 47: “were identified”
  • Line 74: “Drugs that have proven to” instead of “which are proved” (repeated in line 153)
  • Line 94-95: confusing
  • Line 105: during this step
  • Line 125: the same phenomenon
  • Line 137: Src-mediatated (confusing)
  • Line 198: “to whichHSD11b1 catalyzes from cortisone..”??
  • Line 227-229: confusing
  • Line 233-235: confusing

Concerning the content:

  • I suggest to change the title, since the title sounds like a research article in which a screening platform has been generated that identified an anti-metastatic drug. I prefer more precise titles such as “Zebrafish-based screening models employed in the search for inhibitors of processes involved in metastasis.”
  • I would suggest to add the full name when a new abbreviation is introduced (e.g. CXCR4 (line 120), EMT (line 156), VEGF (line 210), MYC, PMEL, TYRP1, TRY (all line 290))
  • I also suggest to add a short note on the transgenic fish line when the specific fish line is introduced first (e.g. first mentioning of TG(Fli1:EGFP) – fluorescence labelled blood vessels)
  • Line 40-41: Give a short explanation of the processes
  • Line 68: add that the procedure that you describe explains the xenograft model
  • Line 70/71: change to “days post fertilization (dpf)”
  • Line 73: does metastasis not include all these processes? That sounds odd to me.
  • Line 76: Table 1: The compounds should follow the order that they are mentioned in in the text. 
  • Line 78: Table 1: Row 4: “cancer cell line” instead of “cancer cell”
  • Line 94-95: The sentence does not contain any information and is confusing, I would remove it from the manuscript.
  • Line 95: I think this refers to Figure 1 instead of Figure 2
  • Line 98: I would add a “treated” and “untreated” text to the figure
  • Line 115 and 117: “vehicle-treated fish” instead of “the fish” - in general, I would not talk about the fish but rather specify e.g. zebrafish larvae
  • Line 118-119: “show” instead of “confirm”, also add “confirming the target” at the end of the sentence
  • Line 129: “In addition” instead of “And”
  • Line 132: “, and so does the genetic inhibiton of CXCR4”
  • Line 146: inhibitors
  • Line 207: Change to: The growth potential of avascular tumors is severely restricted because of the lack of a blood supply.
  • Line 218: I think its “were” instead of “are”, since the study you refer to happened already in 2002. Also, add references that used this platform
  • Line 221: “In vivo” instead of “in vivo”
  • Line 222: “this zebrafish line” instead of “the fish”, “identified” instead of “identify”
  • Line 233: DsRed-labelled cells and GFP-labelled vessels?
  • Line 235: Mention that MDA-435 overexpresses RhoC (something along the lines of: the MDA-435 cells stably over-express RhoC, a small signaling G-protein whose overexpression is associated with...)
  • Line 255 and 258: In vivo italicize
  • Line 258: Remove “And”
  • Line 271: “many other cell types” instead of “other type of cells”
  • Line 276: Add “The” at the beginning of the sentence
  • Line 281: demonstrated that the interaction…
  • Line 290: The ZMEL1 cells that are transplanted into EDN3-deficient zebrafish decrease the metastatic tumor formation compared to the wild type.

Author Response

    I suggest to change the title, since the title sounds like a research article in which a screening platform has been generated that identified an anti-metastatic drug. I prefer more precise titles such as “Zebrafish-based screening models employed in the search for inhibitors of processes involved in metastasis.”

We change the title. New title is " Zebrafish-based screening models for the identification of anti-metastatic drugs".

    I would suggest to add the full name when a new abbreviation is introduced (e.g. CXCR4 (line 120), EMT (line 156), VEGF (line 210), MYC, PMEL, TYRP1, TRY (all line 290))

We improved the points.

    I also suggest to add a short note on the transgenic fish line when the specific fish line is introduced first (e.g. first mentioning of TG(Fli1:EGFP) – fluorescence labelled blood vessels)

We added a short note in following transgenic zebrafish.

Tg(fli1:EGFP), which expresses EGFP in all blood vessels

Tg(kdrl:EGFP), which expresses EGFP in endothelial cells

Tg (lyve1:EGFP) which expresses EGFP in the lymphatic vessels.

Tg (mitfa:BRAFV600E; p53-/- ) fish which develops melanoma by 4 months from their birth

    Line 40-41: Give a short explanation of the processes

We added short explanation in each of the processes.

Metastasis consists of a multi-step process: invasion in which neoplastic epithelial cells invade into the adjacent tissue after they lose cell-cell adhesion; intravasation in which tumor cells penetrate through the endothelium of blood or lymphatic vessels to enter the systemic circulation; survival in the circulatory system in which certain circulating tumor cells appear able to survive in the bloodstream; extravasation in which cancer cells extravasate through the capillary endothelium at distal sites; colonization in which cancer cells proliferate in a new environment, and metastatic tumor formation in which cancer cells form a new tumor in secondary organs (

    Line 68: add that the procedure that you describe explains the xenograft model

    Line 70/71: change to “days post fertilization (dpf)”

We changed from days post-their birth to days post fertilization (dpf)

    Line 73: does metastasis not include all these processes? That sounds odd to me.

We re-wrote this part. Zebrafish models of metastasis does not cover all of step of metastasis. They model the processes of invasion, migration, extravasation, and angiogenesis.

    Line 76: Table 1: The compounds should follow the order that they are mentioned in in the text.

We modified the table.

    Line 78: Table 1: Row 4: “cancer cell line” instead of “cancer cell”

We modified this point.

    Line 94-95: The sentence does not contain any information and is confusing, I would remove it from the manuscript.

We removed the sentence below. "Each of the models has different advantage and offers different approach for that".

    Line 95: I think this refers to Figure 1 instead of Figure 2

This point is an error. We changed from figure2 to Figure1.

    Line 98: I would add a “treated” and “untreated” text to the figure

In this experiment, untreated group does not exist. All of zebrafish were treated with either DMSO as vehicle or Adrenosterone. Each of figures labelled with either vehicle or Adrenosterone.

    Line 115 and 117: “vehicle-treated fish” instead of “the fish” - in general, I would not talk about the fish but rather specify e.g. zebrafish larvae

We changed this point.

    Line 118-119: “show” instead of “confirm”, also add “confirming the target” at the end of the sentence

We changed this sentence. Genetic inhibition of αv integrin shows the same effect and validates that the metastasis suppressing effect of GLPG0187 result from inhibition of αv integrin not off-target effect of it.

    Line 129: “In addition” instead of “And”

We changed this point.

    Line 132: “, and so does the genetic inhibiton of CXCR4”

We corrected this point.

    Line 146: inhibitors

We corrected this point.

    Line 207: Change to: The growth potential of avascular tumors is severely restricted because of the lack of a blood supply.

We changed this sentence.

    Line 218: I think its “were” instead of “are”, since the study you refer to happened already in 2002. Also, add references that used this platform

We changed from "are" into "were". And we added several references in this sentence.

    Line 221: “In vivo” instead of “in vivo”

We corrected this point.

    Line 222: “this zebrafish line” instead of “the fish”, “identified” instead of “identify”

We corrected these points.

    Line 233: DsRed-labelled cells and GFP-labelled vessels?

We changed to "the DsRed-labelled cells and the GFP-labelled vessels"

    Line 235: Mention that MDA-435 overexpresses RhoC (something along the lines of: the MDA-435 cells stably over-express RhoC, a small signaling G-protein whose overexpression is associated with...)

    Line 255 and 258: In vivo italicize

We corrected this point.

    Line 258: Remove “And”

We corrected this point.

    Line 271: “many other cell types” instead of “other type of cells”

We change from “other type of cells” to "many different cell types"

    Line 276: Add “The” at the beginning of the sentence

We corrected this point.

    Line 281: demonstrated that the interaction…

We corrected this point.

    Line 290: The ZMEL1 cells that are transplanted into EDN3-deficient zebrafish decrease the metastatic tumor formation compared to the wild type.

We corrected this point.

Reviewer 2 Report

The review by Nakayama and Makinoshima is focused on the current status of Zebrafish chemical screens for anti-metastasis drugs. Here, they discuss the utility of zebrafish as a system for in vivo screens and describe the number of successful outcomes with respect to compounds that show efficacy in these assays.

The review is comprehensive and the tables highlight the number of published screens with key findings. The main focus is on screens that use transplantation of human cancer cells into the zebrafish embryos with assay endpoints that address cancer cell migration. Another phenotypic assay is to quantify angiogenesis and chemicals that block this process. The thinking is that these will be useful to suppress metastatic cancer cells from seeding a new site of tumor mass.

Overall this is review does an excellent job of covering many published screens and this would be of interest to readers in the field of cancer.

There are a number of comments that should be addressed prior to publication.

1) The review paints an overly positive picture of anti-metastatic screens in zebrafish. The negatives are not discussed at all and so the authors should highlight limitations with using zebrafish. Examples include: zebrafish embryos prefer 28oC, while human and mouse cancer cells prefer 37oC. The fact that temperature has to be lowered could impact the how these drugs could behave.

2) Another limitation with in vivo chemical screens is that the number of compounds screened are much smaller. The authors should include in their tables total number of compounds screened.

3) The authors should highlight drugs that have come out of these screens that were not discovered through traditional in vitro cancer screens. Most of the drugs/compounds highlighted were predicted to have anti-cancer activity so it would be useful for the authors to highlight how zebrafish screens identified new agents that were not possible with standard in vitro screens, if such discoveries exisit.

4) The review needs a summary/conclusion paragraph that highlights areas they discussed and also what the authors opinion of future directives.

5) The authors state that innate and adaptive immunity are not developed until 21 days post birth (line 68-71). I am not sure if this is correct. Macrophages and Neutrophils are present and circulating in the early embryo. Please check that this statement is true.

6) Throughout the text there are grammar and sentences that are not clear. The authors need to carefully read the document again.

Author Response

1) The review paints an overly positive picture of anti-metastatic screens in zebrafish. The negatives are not discussed at all and so the authors should highlight limitations with using zebrafish. Examples include: zebrafish embryos prefer 28oC, while human and mouse cancer cells prefer 37oC. The fact that temperature has to be lowered could impact the how these drugs could behave.

We added disadvantages of zebrafish model in the draft.

The other limitation associated with zebrafish is that the optimum temperature for breeding zebrafish is 28°C; this temperature differs from that of the human body by 9 °C (37 °C). In xenograft experiments that involve the transplantation of human or mouse cells into zebrafish, zebrafish is maintained at 31-34°C since 28 °C is not optimal for the growth of the transplanted cells. However, previous studies demonstrate that human cancer cells that are inoculated into zebrafish embryos have a better proliferation index at 36 °C than at 34 °C. Therefore, 36 °C is considered the most suitable temperature for testing chemotherapeutic drugs such as 5-fluorouracil. Although zebrafish is a poikilothermic and eurythermal animal, there was a 10% increase in the mortality rate of zebrafish maintained at 36 °C compared to that of zebrafish at 34 °C

2) Another limitation with in vivo chemical screens is that the number of compounds screened are much smaller. The authors should include in their tables total number of compounds screened.

We added total number of the chemicals that are subjected to screening in table 2.

We did not add the number in Table 1 since in vivo chemical screens were not conducted in zebrafish xenograft models

3) The authors should highlight drugs that have come out of these screens that were not discovered through traditional in vitro cancer screens. Most of the drugs/compounds highlighted were predicted to have anti-cancer activity so it would be useful for the authors to highlight how zebrafish screens identified new agents that were not possible with standard in vitro screens, if such discoveries exist.

We added highlight comments in reference part.

4) The review needs a summary/conclusion paragraph that highlights areas they discussed and also what the authors opinion of future directives.

We added a conclusion paragraph in the end of each of section.

5) The authors state that innate and adaptive immunity are not developed until 21 days post birth (line 68-71). I am not sure if this is correct. Macrophages and Neutrophils are present and circulating in the early embryo. Please check that this statement is true.

I deleted the word "innate" from this sentence. I apologize for giving rise to misunderstanding in the sentence. The sentence states that zebrafish embryos have not completely developed their innate and adaptive immune system until 21 days post fertilization (dpf). The state means that zebrafish begins to develop innate immune and adaptive immune system before the onset of blood circulation, and then finishes to develop the system after 21 days dpf. So, before 21 days post fertilization, immune cells which work in innate immune and adaptive immune system, exist in zebrafish. Even in the zebrafish embryo, analogous early macrophages were detected. They are found to differentiate in the yolk sac before the onset of blood circulation, and from there to invade cephalic mesenchyme, followed by brain, retina, and epidermis, in a pattern remarkably similar to that described in mammals and birds. This early macrophage is demonstrated to function as innate immune system. When massive amounts of live gram þ (B. subtilis) or gram  (E. coli) bacteria are injected in the bloodstream of zebrafish embryos, they are quickly phagocytosed and killed by the early macrophages (Herbomel et al., Development 1999).

6) Throughout the text there are grammar and sentences that are not clear. The authors need to carefully read the document again.

We subjected this manuscript into English proofing service.

Reviewer 3 Report

The review by Joji et al., describe the recent advances in the field of zebrafish durg screening for anti-metastasis compounds. Overall, the paper is well-written and worthy pubulishing, and there are some of my comments. 1) The whole review lacks a nice conclusion/summary paragraph. 2) As we have also done some work on the zebrafish xenograft model, we were always confused by the interpretion of the dissemination  behaviors of the xenografted cells. We thought it will depends on he size of the cell, if the cells are big, it may suffocate the blood vessels; if the cells are small, the cells are passively disseminated, and are not regulated much by chemotactic factors. In that case, whether the cells are still circulating or colonized may explain more of the cell behavior. Let me know what you think. 3) In page 4, line 59, "Chemical treatment is simply done 59 through adding the chemicals into the water", it was true but needs to be reminded that zebrafish tumor models were also capable to be used for testing hydrophobic(water-insoluble) compounds, e.g. in PMID 30065049, the authors micro-injected the free fatty acids into the yolk sac(full of cholesterol) of the larvae HCC models for a small-scale screening purpose.

Author Response

1) The whole review lacks a nice conclusion/summary paragraph.

We added a conclusion paragraph in the end of each of section.

2) As we have also done some work on the zebrafish xenograft model, we were always confused by the interpretion of the dissemination  behaviors of the xenografted cells. We thought it will depends on he size of the cell, if the cells are big, it may suffocate the blood vessels; if the cells are small, the cells are passively disseminated, and are not regulated much by chemotactic factors. In that case, whether the cells are still circulating or colonized may explain more of the cell behavior. Let me know what you think.

I have not investigated whether the size of transplanted cells may affect the ability of metastatic disseminations in zebrafish. Previous study of mine demonstrated genetic inhibition of HSD11b1 suppressed metastatic dissemination of HCCLM3 (highly metastatic hepatic cancer cell line) in zebrafish xenograft model (Nakayama et al., Molecular Cancer Research 2020). The size of HCCLM3 expressing shHSD11b1 is same with that of HCCLM3 expressing shLacZ (shRNA control). And another study of mine demonstrate that overexpression of serotonin receptor (5HT2C) promoted metastatic dissemination of MCF7 (non-metastatic breast cancer cell line) in zebrafish xenograft model (Nakayama et al., submitted). The size of MCF7 expressing 5HT2C is same with that of MCF7 expressing vector control. These results indicate that metastatic disseminations of HCCLM3 and MCF7 cells are affected by genetic changes not by the size of these cells.

3) In page 4, line 59, "Chemical treatment is simply done through adding the chemicals into the water", it was true but needs to be reminded that zebrafish tumor models were also capable to be used for testing hydrophobic(water-insoluble) compounds, e.g. in PMID 30065049, the authors micro-injected the free fatty acids into the yolk sac(full of cholesterol) of the larvae HCC models for a small-scale screening purpose.

We modified this paragraph into that "chemical treatment is simply done through adding the chemicals (soluble) into the water. Hydrophobic chemicals are exceptionally injected into the zebrafish by microinjection.".

And we mentioned hydrophobic chemicals in in vivo drug screening using zebrafish in the part which explains disadvantage of zebrafish models.

"A study which screens 23 drugs known to cause cardiotoxicity in humans resulted in 4 of 5 false negative results in zebrafish due to poor absorption. Drug efficacy of false negatives was confirmed by microinjection {DJ., 2003 #155}."

Round 2

Reviewer 1 Report

I think that the changes made based on the reviewers's comments improved the manuscript and therefore, I would like to recommend the manuscript for publication. Even though I chose "accept in present form", I spotted three minor errors that could be removed in the proofs: 

  • L37: "end result" instead of "end of result"
  • L66: Reference formatting is off 
  • L68: "generate" instead of "generates"

Author Response

I think that the changes made based on the reviewers's comments improved the manuscript and therefore, I would like to recommend the manuscript for publication. Even though I chose "accept in present form", I spotted three minor errors that could be removed in the proofs:

    L37: "end result" instead of "end of result"

I modified this point.

    L66: Reference formatting is off

I am not sure what should I do. This sentence " These advantages have made zebrafish a popular platform for drug screening." does not have any reference.

    L68: "generate" instead of "generates"

I modified this point.

Reviewer 2 Report

In this resubmission of the review the authors have addressed the comments I outlined. 

There are a couple of sentences that require clarification. 

Line 72-74: "One limitation of zebrafish is that they are not mammals. Further information obtained from zebrafish has no direct translational significance."

This statement about direct translation is not entirely clear. While zebrafish are not mammals, it does not mean that findings can not be directly translated. For example studies on the use of Thalidomide in pregnant rodents do not cause limb defects, but when used in humans it had a devastating effect on limb outgrowth. It turns out that had the investigators used Thalidomide in zebrafish larvae, they would have noted that fin outgrowth was suppressed and would have labelled this drug as a teratogen. The authors should focus on the main disadvantage for zebrafish in human tumor xenograft chemical screens is the difference in optimal temperatures for growth. 

Line 376-379: This last sentence mis-spells CRISPR as CRISPER. This sentence is confusing as the whole review did not mention Cas9 mediated genome editing so in this context it is unclear as to how this fits in with xenograft metastatic cancer cell lines. This sentence should be removed.  

Author Response

Line 72-74: "One limitation of zebrafish is that they are not mammals. Further information obtained from zebrafish has no direct translational significance."

This statement about direct translation is not entirely clear. While zebrafish are not mammals, it does not mean that findings can not be directly translated. For example studies on the use of Thalidomide in pregnant rodents do not cause limb defects, but when used in humans it had a devastating effect on limb outgrowth. It turns out that had the investigators used Thalidomide in zebrafish larvae, they would have noted that fin outgrowth was suppressed and would have labelled this drug as a teratogen. The authors should focus on the main disadvantage for zebrafish in human tumor xenograft chemical screens is the difference in optimal temperatures for growth.

I removed this statement " Further,  information obtained from zebrafish has no highly direct translational significance."

Line 376-379: This last sentence mis-spells CRISPR as CRISPER. This sentence is confusing as the whole review did not mention Cas9 mediated genome editing so in this context it is unclear as to how this fits in with xenograft metastatic cancer cell lines. This sentence should be removed. 

I deleted this sentence" Combining the unique advantages of zebrafish models with modern technologies such as high-resolution imaging and genome editing using the CRISPER-CAS9 system could accelerate the identification of anti-metastatic drugs. "
